# Ki67 Index in Patnaik Grade 2/Kiupel Low-Grade Canine Cutaneous Mast Cell Tumors with Early Lymph Node Metastasis: A Descriptive Study

**DOI:** 10.3390/vetsci10070436

**Published:** 2023-07-05

**Authors:** Marzia Cino, Elisa Maria Gariboldi, Damiano Stefanello, Kevin Pascal Spindler, Erica Ilaria Ferraris, Emanuela Maria Morello, Luca Bertola, Lorella Maniscalco, Marina Martano

**Affiliations:** 1Department of Veterinary Medical Sciences, University of Parma, Strada del Taglio 10, 43126 Parma, Italy; 2Department of Veterinary Medicine and Animal Sciences, University of Milan, Via dell’ Università 1, 26900 Lodi, Italy; 3Department of Veterinary Sciences, University of Torino, Largo Braccini 2, 10095 Grugliasco, Italy

**Keywords:** Ki67, mitotic count, cutaneous mast cell tumor, HN2 lymph node, prognostic markers, canine

## Abstract

**Simple Summary:**

Patnaik grade 2/Kiupel low-grade (G2/LG) represent the majority of diagnosed canine cutaneous mast cell tumors (cMCTs), with a reported incidence of 53.6–57.6%. It has been reported that a variable percentage of dogs (3–17%) with G2/LG die from tumor-related causes; therefore, it could be hypothesized that a subset of G2/LG tumors could manifest an aggressive behavior. Ki67 and mitotic count (MC) are the two most investigated proliferation markers in cMCTs. However, different Ki67 cut-offs have been proposed by previous studies, and have always been assessed in cMCTs of different histological grade in heterogeneous canine populations; moreover, the lymph nodes’ (LNs) metastatic status has never been included. This retrospective study investigated the usefulness of the Ki67 index in predicting cMCT behavior in a homogeneous cohort of G2/LG with HN2 LNs. Thirty-nine dogs with primary G2/LG cMCTs and HN2 LNs treated with surgery alone were enrolled. The results of the present study report that all dogs treated with surgery alone had a good prognosis; no local, nodal or distant metastasis occurred in the timespan considered, and the Ki67 index did not have prognostic impact.

**Abstract:**

Several studies evaluating Ki67 in canine cutaneous mast cell tumors (cMCTs) have reported its prognostic value when tumors of all histological grades are included. This study aims to evaluate whether the Ki67 index has a predictive value in a homogeneous cohort of G2/LG cMCTs with HN2 lymph nodes (LNs) and to describe the clinical outcome. The second goal was to explore the correlation between the Ki67 index and MC. The medical databases of three institutions were retrospectively searched for dogs undergoing surgical treatment for cMCT and LN extirpation, with a histological diagnosis of G2/LG with HN2 LNs. Information about histological margins, MC, Ki67 index, local recurrence, nodal relapse, distant metastasis, de novo cMCT occurrence and date and cause of death were included. A total of 39 cases were identified. None of these developed local and nodal relapse or metastatic distant disease. Median MC was 1 (0–2). Median Ki67 index was 3.5 (0.7–14.3). Ki67 and MC were not significantly correlated. At the end of the study, 32 (82%) dogs were alive, 7 (18%) dogs were dead from unrelated causes and 4 (10.2%) dogs were lost to follow-up. The median ST was not reached, and the mean was 893 days (104–2241 days). Considering the strict inclusion criteria, dogs affected by G2/LG with HN2 LNs treated with surgery alone may have a good oncologic outcome; the Ki67 index does not have prognostic impact.

## 1. Introduction

In the last decade, the number of studies focusing on canine cutaneous mast cell tumors (cMCTs) has exponentially risen, especially those describing different techniques of sentinel lymph node (LN) mapping and prognostic factors [1,2,3,4,5,6,7,8,9]. The recent inclusion of normal-size or non-palpable LN excision as part of the surgical treatment of cMCTs and Weishaar classification of lymph nodes has increased the range of available histopathological prognostic variables [1,2,3,4,5]. Nevertheless, some of the prognostic factors frequently used in the clinical setting need to be updated in light of the Weishaar et al. (2014) classification regarding the neoplastic involvement of lymph nodes [1].

Patnaik Grade 2/Kiupel low grade (G2/LG) represent the majority of diagnosed cMCTs, with a reported incidence of 53.6–57.6% [10,11,12,13,14]. G2/LG MCTs are considered less aggressive than Patnaik Grade 2/Kiupel high grade (G2/HG), with a 1-year survival rate of 94% compared to 46%, respectively [10,11,12,13,14,15]. It has been reported that a variable percentage of dogs (3–17%) with G2/LG die of causes related to the tumor; therefore, it could be hypothesized that a subset of G2/LG has some still unknown intrinsic characteristics that lead to aggressive behavior [10,11,12]. Nevertheless, in the cited papers [10,11,12], the histological metastatic pattern of lymph nodes was not graded according to the Weishaar system, and proliferation markers such as Ki67 or mitotic count (MC) have not been evaluated [1,10,11].

Mitotic count and Ki67 index are the two most investigated proliferation parameters in cMCTs [12,13,14,16,17,18,19,20,21,22,23,24,25,26,27,28]. MC is the count of mitotic figures in an area of 2.37 mm^2^, avoiding areas of hemorrhage, edema, necrosis, and cysts [29]. Ki67 is a nuclear antigen expressed during the late G1, S, M and G2 phases of the cell cycle, and its expression decreases after mitosis [30]. Mitotic count is one of the four criteria used to classify cMCTs into HG or LG according to Kiupel’s classification system, and different studies have reported a significant difference in survival when a threshold of 5 is applied [14,16,17,18,19].

Several studies have found that the Ki67 index has a prognostic value when all histological grades of cMCTs are included [12,16,17,18,19,20,21,22,23,24,25]. However, the literature reviewed provides a plethora of prognostic Ki67 values; thus, the lack of a unique cut-off limits its usefulness in the clinical setting. To the authors’ knowledge, only four studies investigated the prognostic significance of Ki67 exclusively on patients with G2 cMCTs, but the surgical treatment consisted only of the excision of the cMCTs, and lymphadenectomy was never performed or considered [17,20,21,22]. This represents a gap in the knowledge of the behavior of G2/LG cMCTs.

Weishaar et al. (2014) reported that dogs with early and overt nodal involvement (HN2/HN3) had shorter disease-free interval (DFI) and survival time (ST) when compared with dogs with a non- or pre-metastatic nodal involvement (HN0/HN1) [1], although no information was available regarding histological grading according to Kiupel et al. (2011), Ki67 and MC [1]. In addition, a recent paper showed a very good prognosis for dogs with G2/LG cMCTs with HN2 LNs treated without adjuvant chemotherapy, but also this article did not report data about Ki67 and MC [4].

Considering the paucity of information on cMCT with early nodal metastasis and the large variety of Ki67 indices proposed as prognostic cut-off, this study aimed to evaluate whether the Ki67 index had a predicting value in a homogeneous cohort of G2/LG cMCTs with sentinel/regional HN2 LNs and to describe the clinical outcome of this unique group of animals treated with surgery alone. The second goal was to explore the correlation between the Ki67 index and MC. Based on the authors’ experience, it was hypothesized that G2/LG cMCTs with HN2 LNs have a favorable prognosis regardless of Ki67 index, and MC is not correlated with Ki67.

## 2. Materials and Methods

### 2.1. Selection Criteria

The medical databases of three contributing institutions were retrospectively searched for dogs that had undergone surgical treatment for cMCT and lymph node extirpation between 2017 and 2021, with a histological diagnosis of cutaneous mast cell tumor G2/LG with HN2 LNs, either sentinel or regional. Data of client-owned dogs meeting the inclusion criteria were collected. Due to its non-interventional nature, the study did not require ethical approval; however, all clients consented to surgery as a standard procedure for cMCT treatment. The established inclusion criteria were:Both concomitantly curative-intent surgery for cMCT and regional/sentinel lymphadenectomy;Availability of stained histological slides of both cMTC and LNs and formalin-fixed paraffin-embedded tissues for review by the pathologists;Availability of information about status of margins, MC and Ki67 index from the original histopathology report;Ki67 immunostaining performed with the same antibody and protocol [18];No evidence of distant metastasis identified at preoperative thoracic radiographs and ultrasound-guided spleen and liver cytology (stage IV);Absence of concurrent illnesses that could significantly reduce the survival time.

Follow-up information was updated up to 15 May 2023. If multiple LNs were excised, the highest HN value diagnosed was considered.

Dogs that presented recurrence of previously excised cMCT, multiple synchronous cMCTs (stage III), subcutaneous MCT, stage 0 disease and HN3 LNs were excluded from the study. Dogs treated with neoadjuvant and/or adjuvant therapy were also excluded.

### 2.2. Data Collection

Data retrieved included signalment (sex, breed, age and body weight), tumor location (head and neck, trunk (including tail), limbs, digits, inguinal region (including perineal and scrotal)), tumor size (the longest diameter measured by a caliper at the time of presentation), presence of tumor ulceration, regional lymph node (RLN) identified according to Suami et al. (2013) [31], if the RLN was enlarged compared to the contralateral, and complete blood count and serum biochemistry. For the assessment of the sentinel lymphocenters, preoperative and intraoperative mapping techniques, as previously described [7,8], were also recorded.

Information about surgery date, histological margins, MC, Ki67 index, date and method of detection of local recurrence, nodal relapse, distant metastasis and de novo cMCT occurrence and date and cause of death were included.

Follow-up time was defined as the time from the date of surgery to the date of the last contact with the owner, or last clinical evaluation, or death, or the end of the study. Dogs were followed-up every 3 months by clinical evaluation and abdominal ultrasound for the first year and every 6 months from the second year. If indicated, spleen and liver FNA were performed. 

Local relapse was defined as the presence of macroscopic disease at the scar or within 2 cm of the original surgical site [32,33].

Nodal relapse was ascertained by cytological [34] evaluation of the suspected LN; distant relapse was confirmed by cytological diagnosis of the metastatic disease in other visceral organs, such as spleen and liver.

Metastases to RLN/SLN associated with a de novo tumor were not considered as nodal relapse.

Curative-intent surgery was performed with proportional lateral margins in case of tumors smaller than 2 cm, or with 2–3 cm lateral margins for larger nodules. A deep margin of one fascial plane was taken for all tumors.

### 2.3. Histopathological and Immunohistochemical Examination

All cases from the institutions involved were blindly reviewed by two experienced pathologists (L.M., L.B.) to confirm the diagnosis of G2/LG cMCTS and HN2 LNs and to ensure that the evaluation was performed in a uniform manner, in accordance with the most recent and commonly adopted literature for the evaluation of cMCTs [1,15,29,35].

Furthermore, L.M. and L.B., blinded to the clinical and histological data, reviewed MC and Ki67 indices.

Margins were defined as “incomplete” if cells were detected at or within <1 mm of lateral and/or deep edges, “narrow” if neoplastic cells were present within ≤3 mm of the lateral/deep edges and “complete” if lateral and deep margins were >3 mm [32].

Early metastatic LNs (HN2) were defined according to Weishaar et al. (2014) on serial longitudinal LN sections stained with metachromatic staining (toluidine blue) and examined at 400× (Figure 1) [1].

The Ki67 index was calculated according to the method proposed by Vascellari et al. (2013), reviewing photographs of five randomly selected HPFs (400×) on which the positive cells on a total of 500 cells using ImageJ Image software version 1.53 were counted (Figure 2). For each sample, the Ki67 index was expressed as the average number of Ki67 immunostained cells per 100 cells [18].

Mitotic count was calculated as the absolute number of mitoses in 10 consecutive, nonoverlapping HPF of 0.237 mm² at 40× magnification [29]. Areas of highest mitotic activity were selected, while areas characterized by extensive necrosis and apoptotic or pyknotic nuclei were excluded.

### 2.4. Statistical Analysis

Normality of distributions for the numerical variables was assessed with the Shapiro–Wilk test. Continuous variables were expressed as median and range in case of non-normal distribution, and as mean ± standard deviation (SD) in case of normal distribution.

Frequencies are reported for categorical variables.

Spearman’s correlation coefficient was used to assess potential correlation between the Ki67 index and MC. 

The date of surgery was used as an entry point for the calculation of the time to local recurrence (TLR), the time to nodal relapse (TNR), the time to distant relapse (TDR) and the survival time (ST) and disease free interval (DFI).

Only dogs deceased of cMCT-related causes were considered as events; dogs without recurrence or disease progression at the date of the last visit, end of the study or death were censored.

Statistical analysis was performed using a commercially available statistical software package (MedCalc Statistical Software version 19.5.1; Ostend, Belgium).

## 3. Results

A total of 48 cases were identified. One case was excluded because it was a recurrence, seven cases were removed from the study due to the histological re-classification of subcutaneous instead of cutaneous cMCT at the histological review; H&E-stained slides and archival tissue blocks were not available for the revision in one case and it was excluded. Finally, 39 cases fulfilled the inclusion criteria.

The study population consisted of 5 (12.8%) castrated males, 14 (35.9%) sexually intact males, 16 (41%) spayed females and 4 (10.3%) sexually intact females, with a mean age of 7.6 ± 2.4 years at the time of diagnosis. The mean weight was 29.5 ± 11.2 kg. The most represented breeds were Boxer (n = 6, 15.4%), Labrador retriever (n = 6, 15.4%), mixed breed (n = 5, 12.8%), English setter (n = 4, 10.3%) and Golden retriever (n = 3, 7.7%). Twelve (30.8%) other breeds were represented with two dogs each (Beagle, Shar-pei and Weimaraner) and one dog each (French bulldog, Appenzeller Mountain dog, Dogo Argentino, Siberian husky, Alaskan malamute, Maltese, Pinscher, Pitbull, Tosa-Inu). 

The mean size of the cMCTs was 20.8 mm (SD = 11.1); four (10.3%) cMCTs were ulcerated. Fourteen (36%) cMCTS were located on the limbs, eleven (28.2%) on the trunk, seven (18%) on the inguinal region, four (10.3%) on the head and neck and three (7.7%) on the digital region.

The RLN was clinically enlarged in 15.4% (n = 6) of dogs and was normal-sized in 84.6% (n = 33) of dogs.

Preoperative SLN mapping and excision were performed in 33 (84.6%) cases. In the remaining six (15.4%) cases, a regional lymphadenectomy without preoperative mapping was performed. Sentinel lymph node mapping and extirpation were guided by radiopharmaceutical and methylene blue in 20 (60.6%) cases; in one (3%) case, the radiopharmaceutical alone was used.

Twelve cases (36.4%) underwent peritumoral injection of aqueous contrast medium and indirect computed tomography lymphangiography to detect SLNs. In four of these, peritumoral methylene blue dye was injected to guide lymphadenectomy. Information regarding surgical margins was available for all dogs. In 32 cases (82.1%) cMTS were completely excised, in one (2.6%) case, one of the margins was narrow, and in six cases (15.4%), cMCTs were not completely excised. None of the dogs with narrow or infiltrated margins received further treatment.

The overall median follow-up was 750 days (104–2241 days). During this time, none of the 39 dogs developed local or nodal relapse or metastatic distant disease; therefore, time to local relapse (TLR), time to nodal relapse (TNR) and time to distant relapse (TDR) could not be calculated and mean ST was used as a goal for the statistical analysis.

Seven dogs developed one de novo MCT at other locations at 1550, 1307, 857, 804, 473, 428 and 152 days from the primary surgery, respectively; of these, three dogs were surgically treated after a new staging and were still alive at the end of the study after 1867, 1781 and 663 days; two dogs died due to splenic hemangiosarcoma and metastatic soft tissue sarcoma 174 and 1003 days after the second surgery, respectively. In two cases, the owners elected against further staging and treatment and these dogs were still alive at the of the study, after 1007 and 1249 days from the first surgery.

At the end of the study, 82% (n = 32) of dogs were alive and 18% (n = 7) of dogs were dead from unrelated MCT causes. These dogs underwent euthanasia at 1807, 1674, 789, 524, 484, 120 and 104, days for metastatic soft-tissue sarcoma (n = 1), metastatic osteosarcoma (n = 1), splenic hemangiosarcoma (n = 3), right atrial hemangiosarcoma (n = 1) and pulmonic stenosis (n = 1).

Four dogs (10.2%) were lost to follow-up at 2241, 596, 200 and 180 days. All four dogs were alive, and none had local or nodal relapse or metastatic distant disease at the time of the last contact. The median ST of the entire population was not reached, the mean ST was 893 days (range 104–2241 days, SD ± 527.5), the mean ST excluding dogs lost to follow-up (n = 4) and dogs that died of causes unrelated to the MCT (n = 7) was 903 days.

### Histopathology

Median MC was 1 (0–2). Median Ki67 index was 3.5 (0.7–14.3). The Ki67 index and MC were not significantly correlated (r = −0.05; *p* = 0.76). Due to the lack of events, statistical investigations relating to Ki67 index and outcome were precluded.

Results are summarized in Table 1.

## 4. Discussion

MCT is a multifaceted disease which may sometimes behave in an unexpectedly aggressive way, in spite of the absence of negative clinical factors. Therefore, full staging should be advised in many cases, and the relative costs can be relevant to the owner. A better understanding of the prognosticators may help the clinician in deciding when to ask for a more complete staging. To answer this question, the primary aim of this study was to assess the prognostic significance of Ki67 index in a homogeneous cohort of dogs affected by MCTs and to describe the clinical outcome of this population. To do so, the survival time in a group of dogs with G2/LG cMCTs with HN2 LNs treated with surgery alone, was retrospectively analyzed.

To the best of the authors’ knowledge, this is the first study in which the Ki67 index is investigated in cMCTs with the same histologic grade and the same pattern of LN involvement. The impetus for this study was based on the lack of a unique Ki67 cut-off index useful to help the clinician in defining the prognosis for dogs with this subcategory of cMCTs, despite the large variety of Ki67 cut-offs proposed. This group of cMCTs represents a gray zone, where the risk of over- or undertreatment may be real. A suggestion of the possible benefit of chemotherapy treatments in dogs with HN2 LNs was based on Weishaar’s study [1]. Nevertheless, in that study, no difference in disease-free interval and survival time was observed between HN categories. Differences became statistically significant when HN2 cases were merged with HN3. Furthermore, only six HN2 nodes were included, only Patnaik grade was considered, and Ki67 index was not reported [1]. In addition, information about the presence of lymph node enlargement, and the exclusion of visceral metastasis based on spleen and liver cytology was not available [1]. Moreover, even if no survival benefit has been reported in dogs treated with adjuvant systemic chemotherapy after surgical excision of G1-2/LG with HN2 LNs [4], a subset of these cases may have features of high-risk malignancy, such as anatomic location, higher MC and/or Ki67 index, and incomplete excision. In these cases, it could be important to have further standardized parameters to determine which dogs require close monitoring or adjuvant chemotherapy treatment. In addition, the recent Consensus of the Oncology-Pathology Working Group suggested using not only histologic grade but also other clinical and histological markers, such as prognostic factors, since a subset of Kiupel’s LG cMCTs could behave aggressively [13].

In the authors’ experience, dogs with G2/LG with HN2 LNs, which were not receiving adjuvant chemotherapy for various reasons, had a favorable prognosis. This led to the proposal of this retrospective study in which only dogs affected by low-grade MCTs and early metastatic LNs were included.

Different cut-offs of the Ki67 index have been proposed by previous studies [12,16,17,18,19,20,21,22,23,24,25], but they were assessed considering cMCTs of any histological grade in heterogeneous canine populations; moreover, LNs’ metastatic status has never been considered. Consequently, the usefulness of this parameter is still not completely clear, at least for low-grade cMCTs.

The results of the present study report that all included dogs treated with surgery of the primary tumor and regional or sentinel lymphadenectomy without neoadjuvant or adjuvant chemotherapy had a good prognosis, regardless of the completeness of the excision, highlighting the low aggressive behavior of this subcategory, as already reported in the literature [3]. In fact, no dogs developed local and/or nodal relapse, or distant metastasis, and none died from MCT-related disease. This result may be due to the strict inclusion criteria adopted to exclude possible biases, which may hinder a correct interpretation of the outcome, but it also confirms the behavior of this subgroup of MCTs. Moreover, the results suggest that Ki67 evaluation in this subset of cMCTS may not be worthwhile, contrasting the suggestion of the Consensus of the Oncology–Pathology Working Group [13].

In this study, the counting method described by Vascellari et al. [18] was applied. According to that study, Ki67 counts ≥ 10.6 were significantly associated with an increased incidence of cMCT-related mortality. The results of the present study mostly support this finding; in fact, the median Ki67 index was relatively low (3.5%), although two of the evaluated tumors expressed a Ki67 index greater than 10.6. No dogs died due to the cMCT.

In the study of Maglennon et al. (2008), the Ki67 cut-off value of 1.8% proposed by Scase et al. (2006) was confirmed to be associated with a worse prognosis for dogs with G2 cMCTs [20,23]. Additionally, Berlato et al. (2015) showed that dogs with G2 MCTs, MC > 5 and Ki67 > 1.8% had a significantly higher risk of dying from causes related to the cMCTs [16]. In that study, conducted on 49 dogs, the Ki67 index was calculated using the same methodology described by Maglennon et al. (2008) and Scase et al. (2006) [16,20,23]. Both MC and Ki67 index were identically useful for distinguishing high-risk from low-risk MCTs, although the probability of dying of MCT was higher for dogs with increased MC (HR: 15.4 [4.2–56.9]) compared to dogs with increased Ki67 index (HR: 9.8 [2.7–35.7]) [16]. The authors suggested that these markers could be practical to understand which G2 cMCT could benefit from more aggressive treatment, but they did not recommend evaluating proliferation indices in case of LNs metastases, since these cases already have a worse outcome. The Weishaar classification was not yet published and the prognostic role of LNs status was not clarified [16]. In 2018, the same group of authors published a retrospective study on 90 G2 cMCTs, applying the Kiupel classification system, including 82 LG and 8 HG. They reported that seven of eight (87%) dogs with HG/MC > 5 died from causes related to the MCT, but also eleven of the eighty-two (13%) dogs with LG/MC ≤ 5 died. Thus, to improve the sensitivity in finding aggressive disease in this subset of dogs, they split all LG tumors with MC ≤ 5 into two groups, based on the Ki67 index, and they found a higher risk of dying from MCT in dogs with Ki67 > 1.8% [17]. The LNs status was not known; consequently, any comparison with our study cannot be made. The data on Ki67 in the population observed in the present study are in contrast with what was reported above [16,17,20,23], since the median Ki67 index was 3.5%, but no negative events were observed.

In the study by Necova et al. (2021), the authors hypothesized that dogs with G2, MI ≤ 5 and Ki67 index > 1.8%, due to the reported high risk of MCT-related death, could benefit from adjuvant lomustine treatment, and they found a longer survival time compared to previously cited articles. Nevertheless, due to the lack of a control group to assess the benefit of lomustine administration, the regional LNs status assessed only by palpation and ultrasound, and the lack of Kiupel grading system, the authors could not confirm the benefit of chemotherapy [21].

In the present study, a statistical association between MC and Ki67 index was not found. Although it is known that these two parameters may have a nonlinear association, low MI and high Ki67 could be explained by the fact that the Ki67 protein indicates the percentage of cells in the cell cycle and denotes the cells with the capacity to divide. These cells might also stop reproducing or enter apoptosis without going through mitosis [36,37]. Another possible hypothesis of the lack of correlation between the percentage of cells expressing Ki67 and the number of mitoses per field, could be linked to the difference in the choice of the microscopic fields: MC is calculated in the area of the highest mitotic activity (10 consecutive HPF chosen in this area), whereas for Ki67, the HPF fields (5) are selected at random, and not in the areas of greatest mitotic activity, even though this is indeed the technique described by Vascellari et al., 2013 [18,29].

The absence of relapse (local, nodal or distant) and the good outcome observed in the study population could be due to the selection of a subset of dogs with low-risk cMCTs (G2/LG). Since the studies mentioned above [16,17,20,22] did not take into consideration the LNs status, it is not possible to rule out that some of those dogs with G2 tumors actually had metastatic LNs, and their shorter survival could have been due to the undetected nodal metastatic disease, rather than to the high Ki67 index. Moreover, it is reasonable to think that in these studies the Ki67 index was significantly correlated with prognosis because both LG and HG tumors were included in the statistical analysis. This means that, probably, higher Ki67 indices were found in G2/HG, and lower Ki67 indices were related to G2/LG [16,17,20,22]. In addition, the present study included dogs in which the regional or sentinel LNs were always removed, and this surgical approach has been recently reported to have a therapeutic impact [3,4]. A study comparing the outcome of dogs with G2/LG MCT in which the regional/sentinel LNs are extirpated or not could help define this impact.

In this study population, six cases had infiltrated excision margins, but no adjuvant treatments (re-excision, chemotherapy, radiotherapy) were undertaken; local recurrence was not detected during the follow-up time. These results are similar to those previously presented by Smith et al. (2017), who reported a low probability of recurrence in G2 MCTs with low Ki67 index (≤23 Ki67-positive cells/grid), even if the threshold adopted was different from the median value (3.5%) observed in the current study [24].

There are some limitations to this study. The lack of events in the study population may have created a bias in the prognostic information, precluding the possibility to adequately evaluate the oncologic outcome of the HN2 LNs category, as well as the role of the Ki67 index. Nonetheless, the absence of relapse (local, nodal or distant) is not a controllable feature, but these results confirm previous studies in which lymphadenectomy showed a therapeutic impact in LG cMCTs with early metastasis [3], also in the absence of adjuvant chemotherapy [4].

Owing to its retrospective nature, survival time was the only measure of outcome in these cases; since none of the included dogs had nodal or distant relapse, and none of them died from causes related to the cMCT, median disease-free interval was not reached. On the other hand, despite the retrospective nature, the concordance in the preoperative staging and postoperative treatment among the three institutions, the long median follow-up time, and the homogeneity of enrolled cases may provide for better reliability and significance of these results and may represent a point of strength that should be supported by a larger prospective study.

## 5. Conclusions

In conclusion, notwithstanding that the prognosis for cMCTs should be determined by the clinician based on several clinical and histological features, the Ki67 index does not have a prognostic impact in G2/LG cMCTs with HN2 LNs. Considering the strict inclusion criteria, which represent the strength of this study, dogs affected by low-risk cMCTs (G2/LG) and early metastatic (HN2) lymph nodes treated with curative intent surgery alone may have a good oncologic outcome, and the clinical oncologist may not need to ask for further immunohistochemical evaluation (i.e., proliferation parameters) for prognostication. Anyway, further prospective studies with a longer follow-up and a larger cohort of dogs are required to confirm these findings.

## Figures and Tables

**Figure 1 vetsci-10-00436-f001:**
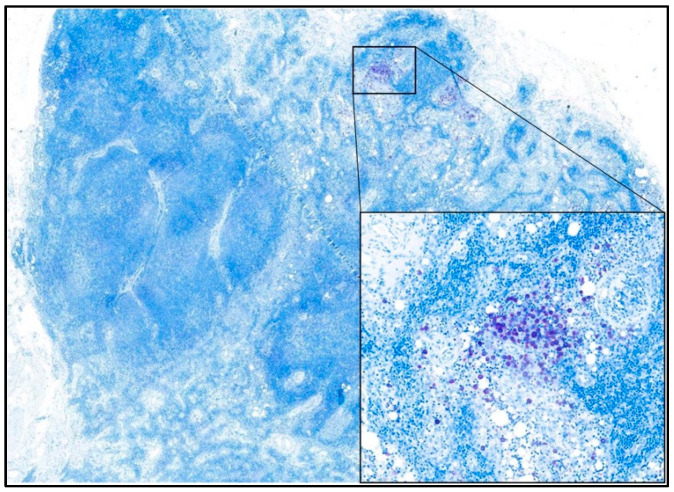
Photomicrograph showing an early metastatic (HN2) lymph node. The toluidine blue staining highlights clusters of neoplastic mast cells without disruption of nodal architecture. In the inset, an aggregate of neoplastic mast cells with abundant intra-cytoplasmic metachromatic granules is visible. 100× and 400× magnification—toluidine blue staining.

**Figure 2 vetsci-10-00436-f002:**
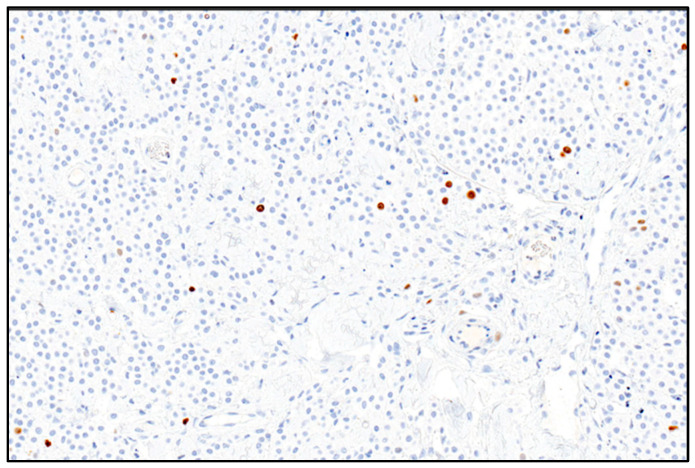
Photomicrograph showing the canine mast cell tumor. Ki67 positive nuclei are marked in brown; negative nuclei are marked in blue and numbered to be counted. 200× magnification—Mayer’s hematoxylin counterstaining.

**Table 1 vetsci-10-00436-t001:** Patients’ demographics and histological details. Abbreviation: F, female intact; FS, female spayed; M, male intact; MN, male neutered; y, years; MCT, mast cell tumor; MC, mitotic count; Y, yes; N, no; Lfu, lost to follow-up; OSA, Osteosarcoma. All dead dogs died from causes not related to the cMCT.

Case	Signalment	MCT Location	Ki67%	MC	Margins	Status	Cause of Death	ST (Days)
1	Labrador retriever, FS, 8 y	Limb	2.5	0	Clean	Dead	Soft tissue sarcoma	1807
2	Golden retriever, FS, 7 y	Limb	1.0	1	Clean	Alive	-	1931
3	Appenzeller Mountain dog, F, 4 y	Limb	1.8	0	Clean	Alive	-	1867
4	Labrador retriever, MN, 9 y	Trunk	1.7	1	Clean	Dead	OSA	789
5	Boxer, M, 8 y	Limb	1.5	1	Clean	Dead	Splenic hemangiosarcoma	1674
6	Weimaraner, MN, 7 y	Limb	3.5	1	Clean	Alive	-	1781
7	Labrador retriever, FS, 6 y	Trunk	0.7	1	Clean	Alive	-	1756
8	Pitbull, M, 3 y	Digit	4.6	1	Clean	Alive	-	1306
9	English setter, M, 8 y	Trunk	1.3	0	Clean	Alive	-	1249
10	English setter, FS, 11 y	Trunk	2.0	0	Clean	Alive	-	1188
11	Maltese, F, 7 y	Head and neck	2.3	0	Dirty	Alive	-	885
12	Tosa Inu, M, 9 y	Inguinal	4.1	0	Clean	Alive	-	801
13	Dogo Argentino	Limb	6.7	1	Clean	Alive	-	760
14	Mixed-breed, FS, 8 y	Limb	5.5	1	Clean	Alive	-	720
15	Golden retriever, FS, 9 y	Limb	3.4	0	Clean	Dead	Splenic hemangiosarcoma	484
16	Mixed-breed, FS, 9 y	Trunk	5.9	1	Clean	Alive	-	690
17	Husky, MN, 9 y	Inguinal	6.1	0	Clean	Alive	-	678
18	Boxer, MN, 8 y	Limb	8.7	0	Dirty	Alive	-	663
19	Labrador retriever, FS, 4.5 y	Inguinal	8.8	0	Clean	Alive	-	613
20	Boxer, M, 8 y	Head and neck	11.5	0	Clean	Dead	Right Atrial hemangiosarcoma	104
21	English setter, FS, 5 y	Trunk	14.3	0	Clean	Alive	-	557
22	Shar-pei, FS, 6 y	Limb	9.1	1	Dirty	Lfu	-	2241
23	French bulldog, MN, 11 y	Digit	5.2	0	Dirty	Dead	Heart failure (by pulmonic stenosis)	120
24	Shar-pei, F, 10 y	Head and neck	1.9	1	Dirty	Lfu	-	596
25	Mixed-breed, FS, 11 y	Limb	8.4	1	Clean	Lfu	-	200
26	Beagle, FS, 13 y	Inguinal	3.0	1	Dirty	Lfu	-	180
27	Pinscher, M, 5 y	Inguinal	2	1	Clean	Alive	-	935
28	Beagle, M, 10 y	Inguinal	3.9	2	Clean	Alive	-	882
29	Boxer, F, 7 y	Limb	2.8	0	Clean	Alive	-	588
30	Golden retriever, M, 7 y	Trunk	1.9	0	Clean	Alive	-	578
31	Labrador retriever, M, 11 y	Head and neck	2.9	0	Clean	Alive	-	1007
32	Mixed-breed, FS, 9 y	Limb	7.4	1	Clean	Dead	Splenic hemangiosarcoma	524
33	Boxer, FS, 5 y	Trunk	8.1	2	Clean	Alive	-	760
34	Alaskan malamute, M, 5 y	Trunk	6.2	1	Clean	Alive	-	792
35	Labrador retriever, M, 3.5 y	Trunk	7	0	Clean	Alive	-	750
36	English setter, M, 10.5 y	Inguinal	3.2	0	Clean	Alive	-	728
37	Mixed-breed, M, 8 y	Digit	4.3	2	Clean but close	Alive	-	539
38	Boxer, M, 6 y	Limb	5.6	1	Clean	Alive	-	544
39	Weimaraner, FS, 7 y	Trunk	3.5	0	Clean	Alive	-	553

## Data Availability

Data are shown in the article.

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
