# Peer review of "Ki67 Index in Patnaik Grade 2/Kiupel Low-Grade Canine Cutaneous Mast Cell Tumors with Early Lymph Node Metastasis: A Descriptive Study"

_vetsci, 2023, doi:10.3390/vetsci10070436_

Round 1

Reviewer 1 Report (Previous Reviewer 1)

The authors have improved the quality of the manuscript, but there are still some points that I suggest reviewing.

-       I would like to see some histological/IHC photomicrographs of the MCTs and LNs, correlation graphs and survival curves;

-       The authors evaluated Ki67 index in a total of 500 cells using ImageJ (Figure 1) and Ki67 index was expressed as the average number of Ki67 immunostained cells per 100 cells. In this image, for example, I can see that almost 200 cells were counted. What happens in the next 4 images you have for this case? I think this is a biased and low reproducible evaluation.

-       According to Vascellari et al. (2013), Ki67 counts ≥ 10.6 were significantly associated with increased MCT-related mortality, in a study that compared MCTs from 3 grades. Since the median Ki67 index in your study was relatively low (3.5%), could you try to establish a particular cut-off for your cases (G2/LG MCTs), in order to evaluate Ki67 counts (that must be improved, as I mentioned in the above comments)?

Minor editing required.

Author Response

Reviewer 2 Report (Previous Reviewer 2)

The authors have greatly improved their text: the grades of each tumour have been reviewed by the same pathologists; and every remark made at the time of the first review has been taken into account, with an appropriate response.

As a result, the cohort is now really extremely homogeneous : all cMCT were Patnaik grade 2 and Kiupel low grade with sentinel or regional pre-metastatic lymph nodes, HN2, with a negative distant liver and spleen cytological examination, with surgery alone (excision of the tumour and the sentinel or regional LN) and without any adjuvant treatment. This cohort is much more homogeneous than the previous ones published (the LN were not always systematically removed surgically ), which gives this cohort a real weight, even if it only contains 39 cases: indeed, the absence of any event could be partly linked to this systematic removal of the LN, which will have to be demonstrated with a larger cohort and prospectively, and indeed the matter of adjuvant therapy in these cases does not seem to be relevant any more, which is very interesting.

Just one comment: apart from the possible apoptosis before mitosis of cMCT cells expressing ki67,  the lack of correlation between the % of cells expressing ki67 and the number of mitoses per field could also be linked to the difference in the choice of the microscopic fields: on the one hand, are chosen, as described elsewhere, the area with the highest number of mitoses (10 consecutive HPF chosen in this area), whereas for ki67, the HPF fields (5) are selected at random, and not in the areas of greatest mitotic activity, even though this is indeed the technique described by Vascellari et al. , 2013.

line 63: in the study (10) LN were not surgically removed and in the 2 other studies (11-12), they were either surgically removed or just evaluated by cytology; it makes a great difference with your current study, which is much more homogeneous.

Line100: "dogs undergone...": better "dogs that undergone..."?

Author Response

Dear reviewer, 

We very grateful for your suggestions. We really appreciate reviewers who try to improve the quality of research/paper with their knowledge.  

I added some of your comments to the manuscript

This manuscript is a resubmission of an earlier submission. The following is a list of the peer review reports and author responses from that submission.

Round 1

Reviewer 1 Report

The flaws regarding the reproducibility of the methods for LN selection and Ki67/MC quantification, as well as inclusion/exclusion criteria (e.g., dogs with other neoplastic or heart diseases) remain within the manuscript. 

Reviewer 2 Report

The manuscript has been substantially amended; Thank you to have added the Table.

In the introduction, the assumption that ki67 and MC values would not be correlated for G2/LG HN2 cMCTs still remains unexplained, although it is counter-intuitive and deserves to be.

It is a pity that the grades were reviewed by a different pathologist from one institute to another, whereas it was specified in the first version that two of the authors had reviewed all the ki67s and MCs in a blinded fashion: why not also have had all the grades reviewed by these two pathologists since then, the study would have definitly gained in strength. The sentence stipulating that the ki67s and MCs were reviewed in a blinded fashion by two of the authors should at least be reinstated in the second version, as it was in the first version.

Line 223 it lacks the word « at the end of… »

Line 228 atrial hemangiosarcoma (not hemagiosarcoma) and pulmonary stenosis not « pulmonic » stenosis

Table 1 case 23 pulmonary stenosis not « pulmonic »